# A Machine-Learning-Based Approach to Prediction of Biogeographic Ancestry within Europe

**DOI:** 10.3390/ijms242015095

**Published:** 2023-10-11

**Authors:** Anna Kloska, Agata Giełczyk, Tomasz Grzybowski, Rafał Płoski, Sylwester M. Kloska, Tomasz Marciniak, Krzysztof Pałczyński, Urszula Rogalla-Ładniak, Boris A. Malyarchuk, Miroslava V. Derenko, Nataša Kovačević-Grujičić, Milena Stevanović, Danijela Drakulić, Slobodan Davidović, Magdalena Spólnicka, Magdalena Zubańska, Marcin Woźniak

**Affiliations:** 1Department of Forensic Medicine, The Ludwik Rydygier Collegium Medicum in Bydgoszcz, Nicolaus Copernicus University in Torun, 85067 Bydgoszcz, Poland; 2Faculty of Medical Sciences, Bydgoszcz University of Science and Technology, 85796 Bydgoszcz, Poland; 3Faculty of Telecommunications, Computer Science and Electrical Engineering, Bydgoszcz University of Science and Technology, 85796 Bydgoszcz, Poland; 4Department of Medical Genetics, Warsaw Medical University, 02106 Warsaw, Poland; 5Institute of Biological Problems of the North, Russian Academy of Sciences, 685000 Magadan, Russia; 6Institute of Molecular Genetics and Genetic Engineering, University of Belgrade, 11042 Belgrade, Serbia; 7Faculty of Biology, University of Belgrade, 11000 Belgrade, Serbia; 8Serbian Academy of Sciences and Arts, 11000 Belgrade, Serbia; 9Institute for Biological Research “Siniša Stanković”, National Institute of Republic of Serbia, University of Belgrade, 11060 Belgrade, Serbia; 10Center of Forensic Sicences, University of Warsaw, 00927 Warsaw, Poland; 11Faculty of Law and Administration, Department of Criminology and Forensic Sciences, University of Warmia and Mazury, 10726 Olsztyn, Poland

**Keywords:** machine learning, SVM, biogeographic origin, biogeographic ancestry

## Abstract

Data obtained with the use of massive parallel sequencing (MPS) can be valuable in population genetics studies. In particular, such data harbor the potential for distinguishing samples from different populations, especially from those coming from adjacent populations of common origin. Machine learning (ML) techniques seem to be especially well suited for analyzing large datasets obtained using MPS. The Slavic populations constitute about a third of the population of Europe and inhabit a large area of the continent, while being relatively closely related in population genetics terms. In this proof-of-concept study, various ML techniques were used to classify DNA samples from Slavic and non-Slavic individuals. The primary objective of this study was to empirically evaluate the feasibility of discerning the genetic provenance of individuals of Slavic descent who exhibit genetic similarity, with the overarching goal of categorizing DNA specimens derived from diverse Slavic population representatives. Raw sequencing data were pre-processed, to obtain a 1200 character-long binary vector. A total of three classifiers were used—Random Forest, Support Vector Machine (SVM), and XGBoost. The most-promising results were obtained using SVM with a linear kernel, with 99.9% accuracy and F1-scores of 0.9846–1.000 for all classes.

## 1. Introduction

Progress in the field of DNA sequencing resulting from the advancement of technologies collectively referred to as next-generation sequencing (NGS) or massive parallel sequencing (MPS) has led to a significant increase in the amount of available genomic data.

MPS technology is typically utilized for molecular diagnostics and to assess a predisposition to genetic diseases or to look for treatment options [1]. However, data obtained using different applications of MPS technology also provide insight into other aspects of the genetics of living organisms, such as the distribution frequency of polymorphic variants. Therefore, these data can be a valuable tool for population and forensic genetics, environmental biology, and related fields, as they contain massive amounts of information that would be hard to collect using other common techniques of molecular biology. In particular, such data could be useful for distinguishing adjacent populations of common origin, which is currently a major challenge [2].

Due to the huge amount of data generated by MPS analysis, it is necessary to use properly configured information technology (IT) systems and algorithms for data handling and analysis. Machine learning (ML) techniques seem to be particularly well-suited to analyzing data obtained using MPS [3]. In recent years, ML has been used across many different disciplines and for many different research tasks, including bioinformatics, image recognition, and predictive DNA analysis. ML techniques have been used with great success in the analysis of the human genome, allowing quick and efficient acquisition of useful data [4,5,6].

One of the most-important advantages of using ML is the fact that data obtained empirically are used as training data, from which the model can learn any present patterns. These patterns are then used for the classification of other datasets. Due to the growing amount of genomic data generated by projects such as The 1000 Genomes Project [7] and The Cancer Genome Atlas (TCGA), genomic databases are growing rapidly. Data from such databases are usually described in detail; thus, they can serve as the reference data on which ML models could learn genetic patterns occurring in particular populations. Unfortunately, many populations are underrepresented in global databases collecting genomic data. For example, in The 1000 Genomes Project, European populations are represented mostly by samples from Western Europe (Great Britain), Southwestern Europe (Italy and Spain), and Finland, while populations living in the southeastern parts of the continent (including Slavic countries) are virtually absent.

Slavs constitute about a third of Europeans and inhabit almost half of the continent’s territory, mainly in its central and eastern parts. Slavic populations are divided based on linguistic criteria into Western Slavs (e.g., Poles, Czechs, Slovaks), Eastern Slavs (e.g., Russians, Ukrainians, Belarusians), and Southern Slavs (e.g., Serbs, Croatians, Slovenians). Until recently, genetic research on the diversity of Slavs has focused on autosomal and Y-chromosomal short tandem repeats (STRs) and mitochondrial DNA (mtDNA) [8,9,10,11,12,13]. Thus far, no polymorphic loci or analytical methods have been developed that would allow the distinction of genetic material from representatives of individual groups of Slavs with high certainty.

The main aim of this research was to test whether it is possible to distinguish the biogeographic origin of Slavic individuals using various ML techniques and to correctly classify DNA samples from representatives of different Slavic populations. In addition to the technical aspect of this analysis, focused on the confirmation of the effectiveness of ML methods in a new application and the indication of the most-effective algorithms, one can also think about the potential practical applications of such a methodology, including:Relatively inexpensive acquisition of population data to search for ancestry-informative markers (AIMs) using data from public databases instead of conducting separate experiments for marker acquisition. Exome sequencing is an analytical technique used more and more widely, so the amount of available exome data from particular populations will increase over time.Inspecting genomic or exome sequencing data sent to public sequence repositories, i.e., verification and correction of possible errors in the sample description or ascribing unknown samples to their populations of origin.It can also be imagined that, in the future, when the MPS methodology becomes less expensive and simpler to use, individual genetic identification (such as in forensic applications) will use data derived from large fragments of the genome. The proposed methodology could be applied to such data.

The increase in the popularity of ML methods has allowed the analysis of DNA sequencing results in a more-complex and much-faster way, which is often impossible to implement using standard analytical workflows. Currently available ML algorithms depend on many parameters, with specific pros and cons [14].

The interest in ML methods is mainly due to the importance of links between nucleotide sequences and disease. For example, Bellot et al. [15] attempted to predict complex human traits based on genetic and phenotypic data from a publicly available human database. This information included features such as body height, bone mineral density, body mass index (BMI), and blood pressure. With the use of neural networks to analyze such a huge amount of data, Bellot et al. [15] confirmed a number of candidate single-nucleotide polymorphisms (SNPs) already suspected to be associated with the examined features. In addition, they managed to select new SNPs potentially affecting the phenotype. Due to the amount of information available in such databases, a manual analysis becomes impossible. Therefore, the use of ML-based bioinformatic tools for this purpose is extremely useful and valuable.

Battey et al. developed a method of biogeographical origin prediction using deep neural networks [16]. Their proposed algorithm allowed the determination of the origin of test samples. The authors ensured that the tool was highly effective and that the analyses were carried out quickly. However, they also indicated that their model could be improved.

Another example of an ML application for genomic sequence analysis is the reconstruction of historical human migrations. The Dutch GoNL project entailed whole-genome sequencing of individuals from 250 families [17]. By determining genetic variants, as well as de novo mutations occurring in genomes of the examined group of people, the researchers were able to map the course of migration of the Dutch population, in connection with such factors as changes in water level, which forced people to leave their current places of residence and settle in new areas. These results were compared with historical data to authenticate the acquired information, and their accuracy was subsequently confirmed and verified.

However, ML methods are not infallible [15,18]. One limitation can be caused by oversampling of the chosen classifier in the event of an excessive amount of input data being used as a training set. An overtrained classifier consequently has problems when analyzing test data, which leads to their misinterpretation. Finding the “golden mean”, to avoid overtraining the classifier and, at the same time, providing it with the right amount of information so that it can serve its purpose can be challenging.

The majority of inherited traits in humans are polygenic [19,20]. Thus, the task required of the ML algorithm is relatively complicated, which can affect the reliability of the results obtained. Nevertheless, ML is one of the options for the analysis and interpretation of large and complex datasets [21,22]. ML is a relatively young field of science that is still developing and has not yet reached its maximum potential, which puts it in an extremely favorable position for future research.

The primary objective of our presented study was to establish a precision-driven methodology for predicting the biogeographic origin of individuals hailing from diverse European countries. To accomplish this objective, we utilized ML-based approaches, enabling an efficient and sophisticated means to tackle this intricate task. Our goal encompassed the development of a resilient and trustworthy framework, poised to aid in the discernment of individuals’ biogeographic ancestry. Such an advancement carries the potential to yield profound insights into European population genetics, thereby fostering applications across disciplines such as anthropology, forensics, and medical research.

In this study, we introduced several novel contributions to the field of biogeographic ancestry prediction. One key aspect of our study is the utilization of a genetic algorithm (GA) in the feature-selection process. Through the GA, we were able to identify the minimum essential set of SNPs required to accurately distinguish the considered populations. This approach not only enhances the efficiency of our models, but also mitigates the risk of overfitting. By selecting only the most-informative genetic markers, we enabled ML models to achieve improved generalization and robustness, making the presented method a valuable addition to the field of biogeographic ancestry prediction within Europe.

## 2. Results

### 2.1. Feature Selection

The results of the GA analysis are presented in Figure 1. According to the obtained histogram, the minimum number of SNPs sufficient to distinguish the studied populations was 300. In fact, an increase in the number of SNPs turned out to be optimal for our data as increasing the number of SNPs considered by the GA gave lower minimum F1-score values for the best model. The list of 300 SNPs providing the highest minimum F1-score is presented in Appendix A.

### 2.2. Method’s Evaluation

The results of the analyses performed are presented in Table 1. The presented values are the mean of the results obtained in each fold calculated using the aforementioned Equations (Equation 1) and (Equation 2). The metrics, including the accuracy and the values of the F1-score, were calculated based on the confusion matrix generated during the evaluation process (Figure 2). In particular, it is important to note that the parameters were calculated in a “1 vs. all” fashion. In the context of our study, this means that, when TP represented correctly classified Poles, TN represented the sum of correctly classified samples from the non-Polish ancestry groups, such as Russian, Serbian, and non-Slavic populations. As per the results in Table 1, linear SVM provided the most-promising results, with an accuracy of 0.9990 and an F1-score ranging from 0.9846 to 1.00, depending on the class.

The presented results varied for each classifier tested, with accuracies of 0.978–0.999. These results showed no significant differences. However, larger differences could be seen in the F1-scores. The F1-score for the Serbian population was significantly lower than that of other populations, regardless of the classifier used. The lowest overall F1-score was that for the Serbian population classified using a Random Forest approach. This observation may be due to the small number of samples of this origin (n = 20). All kernels of the SVM provided comparable results apart from the Serbs class. For this class, the F1-score was between 0.6267 and 0.9846, with the polynomial SVM the poorest performing and the linear SVM the best performing.

## 3. Discussion

Allocco et al. [23] developed a method for predicting geographic origin based on Naive Bayes. Naive Bayes is a simple predictive algorithm, which, thanks to its simplicity and speed of operation, could be used to select potential SNPs that would allow the authors to distinguish the origins of those individuals tested. The accuracy of this method was approximately 95% across 50 randomly selected SNPs for each class. However, it is worth noting that this method was developed with a cohort whose origin differed significantly compared to our subjects, as the authors included samples of African, Asian, and European origin. Due to the greater differences in the genotypes of individuals from such distant regions, the classification of this cohort was much easier than that of the group of Slavs we studied, whose genomes were more similar. For this reason, our model compared more SNPs (300 vs. 50). Our emphasis on the number of geographic origins involved in the study underscores the complexities and unique genetic landscape presented by different population groups. By focusing on the Slavic ethnic group, our model accounted for the subtle genetic variations, ensuring meaningful differentiation and classification within this specific cohort. While both studies employed evaluation metrics, we aimed to highlight the importance of contextualizing accuracy percentages in light of the specific research objectives and genetic diversity of the studied populations. Our intention was not to directly compare these metrics, but rather to emphasize the significance of our method’s efficiency in distinguishing similar populations with reduced genetic disparities.

Compared to the method used by Battey et al. [16], our method is faster to carry out and requires less computing power. Their “Locator” method is based on deep neural networks (DNNs) used to predict the geographic origin of a sample from its genotype. In that method, 100,000 SNPs are used for the calculations, which can lead to problems with the RAM of the computing machine. The use of 300 SNPs requires significantly less computing power, which speeds up the classification.

The “SPASIBA” method proposed by Guillot et al. [24] uses genetic datasets of known geographic origin as a “baseline”. For each population, site-specific reference alleles are selected. Samples of unknown geographical origin are compared and classified to the appropriate location. Using this method, tens of thousands of SNPs are compared, and the results can be obtained within several minutes. However, the methods described by Battey et al. [16] and by Guillot et al. [24], due to their complexity, are better suited to define geographical origin in the context of ancestral migration, etc.

## 4. Materials and Methods

The proposed workflow is presented in Figure 3. First of all, the data were obtained by the MPS technology. Then, the pre-processing was performed to convert the information from DNA sequencing to the form readable by the ML techniques. The subsequent phase, denoted as feature selection, was implemented to mitigate model complexity and streamline computational efficiency. The ultimate stage of the processing pipeline encompassed classification, wherein an ML-based approach was employed to discern the origin, specifically distinguishing between Poles, Serbs, Russians, and non-Slavs.

### 4.1. Data Acquisition

The Slavic populations’ (Polish, Russian, and Serbian) data used for this research were obtained as part of the NEXT project (DOB-BIO7/17/01/2015). The NEXT project was an R&D scientific project aimed at constructing a tool for the differentiation of DNA samples coming from different Slavic populations based on a selection of SNP loci. DNA samples were collected from randomly selected individuals representing particular populations. Polish, Russian, and Serbian populations were selected for the discovery phase of the project to represent, respectively, Western, Eastern, and Southern Slavs. The collected samples were subjected to exome sequencing, and the resulting data were used in the search for population-specific polymorphisms that were included in the SNP set chosen for further analyses in the project. Blood samples were obtained with written consent from 127 Poles, 29 Russians, and 20 Serbs (approval by the Ethics Committee of the Jagiellonian University in Krakow, Decision No. KBET/122/6120/11/2016, and by the Ethics Committee of the Institute of Molecular Genetics and Genetic Engineering, the University of Belgrade (Decision No. O-EO-005/2016)). High-quality genomic DNA extracted from peripheral blood was used for library preparation using SeqCap EZ MedExome Plus with custom probes’ spike-in-captured space of 1,310,077 bp (Roche Diagnostics GmbH, Mannheim, Germany). The full target list is presented in Appendix A. Libraries were prepared according to the manufacturer’s recommendations and paired-end sequenced (2 × 100 bp) on HiSeq 1500 (Illumina, San Diego, CA, USA). Raw data were analyzed with bcl2fastq v. 1.8.4 software (Illumina) to generate reads in FASTQ format. Raw data (FASTQ files) were fed into a bioinformatics pipeline (CoVaCS [25]), aimed at the analysis of SNPs and indel variations using the GRCh37/hg19 reference human genome sequence [16]. The resulting VCF files contained information on SNPs and indels detected in individual samples, confirmed by at least two out of three independent variant-detection tools (GATK [26], FreeBayes [27], and VarScan [28]). Data for the non-Slavic populations were obtained from The 1000 Genomes Project Database as Phase 3 VCF files, downloaded from a publicly accessible FTP server (https://ftp.1000genomes.ebi.ac.uk/, accessed on 6 October 2023).

Data for the Polish, Russian, and Serbian populations, as well as European samples from The 1000 Genomes Project (British in England and Scotland, Finnish in Finland, Iberian populations in Spain, Toscani in Italy, Utah residents with Northern and Western European ancestry) were used for the study. In total, 678 samples were analyzed. The assignment of these sequences to various populations is presented in Figure 4. Data collected within the NEXT project are available to the public at Mendeley Data (https://data.mendeley.com/datasets/9ffz69hbcf, accessed on 6 October 2023).

### 4.2. Data Pre-Processing

The raw DNA sequences required data pre-processing prior to their use in the ML analysis, in order to ensure the correct functioning of these systems. All sequence data were stored in VCF files in the form of records in which each line contains information as to the chromosome, position, reference, and alternative alleles, as well as a genotype. To render these data amenable to algorithmic interpretation, the VCF files underwent a transformation that translated each polymorphism into a binary vector representation. The pre-processing schema is presented in Figure 5. The blue and orange tables in Figure 5 represent lists of polymorphisms identified among the samples. The green table represents the input dataset for the ML-based methods, which contains polymorphism data for all analyzed samples. For example, consider the instance of “1:13302 C/T”: it holds a value of True (1) for Sample Number 1, indicating the presence of this particular polymorphism in the dataset for that sample. Conversely, for sample n, this element assumes a value of False (0), indicating the absence of the mentioned polymorphism in the data pertaining to sample n. During further sequencing analysis, one of the samples collected from the Russian population was excluded due to the low quality of the obtained sequencing results.

### 4.3. Feature Selection

Given the substantial volume of genomic data procured from each individual sample, additional measures were implemented to curtail the requisite count of indispensable SNPs essential for discerning the specific populations under consideration. A GA [29,30] was employed after initially preparing a histogram that calculated the distinctiveness of each SNP, representing the percentage of individuals within a given population possessing that particular SNP, as well as the total count of individuals across the entire study cohort who shared the SNP. From this list, random SNPs were selected and evaluated using the GA to determine the minimum threshold of SNPs required to achieve highly accurate results. This iterative process served to prevent overfitting of the ML models and significantly contributed to enhancing the robustness of their generalization capabilities.

### 4.4. Classification

All experiments were performed in Jupyter Notebooks, using Python kernels and the Scikit-learn library. These tools enabled the use of ML methods. A total of 3 classifiers were used in this research: Random Forest, XGBoost, and SVM.

Random Forest is an ensemble learning method widely employed in ML for both classification and regression tasks. It is a robust and versatile algorithm that combines the power of multiple decision trees to achieve higher predictive accuracy and generalization performance. In this study, we configured Random Forest with the following parameters: n_estimators = 100, criterion = “gini”, max_depth = None, min_samples_split = 2, and min_samples_leaf = 1. These parameter settings were chosen to maintain the algorithm’s default behavior and ensure a fair comparison with other methods.

Extreme Gradient Boosting (XGBoost) is an advanced ensemble learning technique that has gained immense popularity due to its superior performance in various ML competitions and real-world applications. XGBoost is specifically designed to optimize efficiency and predictive accuracy by employing a gradient boosting framework. In this study, we configured XGBoost with the following parameters: booster = “gbtree”, learning_rate = 0.3, max_depth = 6, lambda = 1, alpha = 0, and min_child_weight = 1. These parameter settings were chosen to align with common default configurations and maintain consistency across our experiments.

Support Vector Machine (SVM) is a powerful and widely used supervised ML algorithm for both classification and regression tasks. Its core principle is to find an optimal hyperplane that best separates data points belonging to different classes in a high-dimensional feature space. In order to optimize the method, four different kernels were tested for SVM—linear, polynomial, RBF, and sigmoid. The distinctions among these SVM kernels are predicated upon their capacity to address distinct types of data, specifically distinguishing between linearly separable and non-linearly separable datasets, the level of complexity in constructing decision boundaries, as well as the utilization of parameters to govern the shape and behavior of these boundaries. The selection of an appropriate kernel is contingent upon the intrinsic characteristics of the data and the particular problem being addressed. Additionally, we set the following parameters for SVM: C = 1, gamma = “scale”, and retained the remaining parameters at their default values for consistent and fair evaluation across different kernel functions.

After the pre-processing stage, the data were stored as binary vectors with lengths of 1200 characters. The number of 1200 elements came from the minimal number of SNPs, which was 300 (detailed information in the Section 2) times all four classes included in the research. These data were split into training (70%) and testing (30%) datasets, using a five-fold cross-validation. This ensured that the obtained results did not depend on the training data. All chosen classifiers were then trained on the training dataset and validated with the test dataset.

### 4.5. Method’s Evaluation

For the evaluation, two parameters were used—accuracy and F1-score—which were based on the confusion matrix, with four outcomes as follows: TP—true positive—a Slavic sample classified as Slavic; TN—true negative—non-Slavic sample classified as non-Slavic; FP—false positive—a non-Slavic sample classified as Slavic; and finally, FN—false negatives—a Slavic sample classified as non-Slavic. These parameters can be expressed with the following Equations (Equation 1) and (Equation 2).
(1)$$\begin{array}{c} Accuracy=\frac{TP+TN}{TP+TN+FP+FN} \end{array}$$
(2)$$\begin{array}{c} F1-score=\frac{TP}{TP+0.5\cdot (FP+FN)} \end{array}$$

## 5. Conclusions

As seen in Table 1, all classifiers were characterized by relatively high accuracy scores, with the lowest value (0.9776) achieved for Random Forest and the highest value of 0.999 achieved for linear SVM. For all classifiers used, very high (>0.99) F1-scores were obtained for non-Slavic and Russian populations, suggesting that all classifiers were able to differentiate samples from those populations in our dataset. The F1-score results were also relatively high for Poles, with the maximum value higher than 0.99 reached by the linear SVM classifier. For Serbs, the F1-score values were much more diverse, spanning the range between 0.397 (Random Forest) and 0.98 (linear SVM).

It follows from these results that our approach allows for high-confidence differentiation between Slavs and non-Slavic populations, as well as between different Slavic populations using exome sequencing data, with the linear SVM classifier providing the highest confidence, both in the accuracy and F1-score.

Despite the high accuracy values obtained for our dataset, it seems that the F1-scores are better estimators of the method’s potential as the F1-score parameter takes into account the FP and FN results. For this reason, the F1-score seems to be more useful for studies of sample sets containing classes differing in size.

### 5.1. Limitations

The high degree of class imbalance within our dataset, as visualized in Figure 5, poses a limitation to our methodology. This class imbalance could have impacted the accuracy of our origin prediction results. While there are strategies available to address class imbalance, such as reducing the size of the majority class or oversampling the minority class through resampling techniques, we chose not to balance the dataset in this particular case. Balancing by reducing the number of data points from Polish and non-Slavic samples would significantly shrink the dataset, potentially rendering it unsuitable for effective machine learning techniques. Conversely, data augmentation methods would introduce quasi-artificial data into the analysis, a practice we sought to avoid.

To enhance the robustness of our methodology and mitigate the impact of class imbalance, it would be beneficial to collect additional samples from the underrepresented classes, explore new populations in the investigation, and increase the sample size for each studied population. Such an approach may potentially reduce the required number of SNPs, subsequently streamlining the computational complexity and expediting the calculation processes.

### 5.2. Future Work

The pipeline presented in this paper was prepared as a proof-of-concept and, thus, is subject to optimization. The experiments may be repeated with different datasets and with other ML classifiers. It could be also valuable to use neural networks, which can provide very promising results when faced with numerous classification issues. The potential extension of the proposed method would be the addition of an explaining module to the classifier. Thus, the method would be able to predict the origin with high accuracy, but also to explain why it made such a decision.

## Figures and Tables

**Figure 1 ijms-24-15095-f001:**
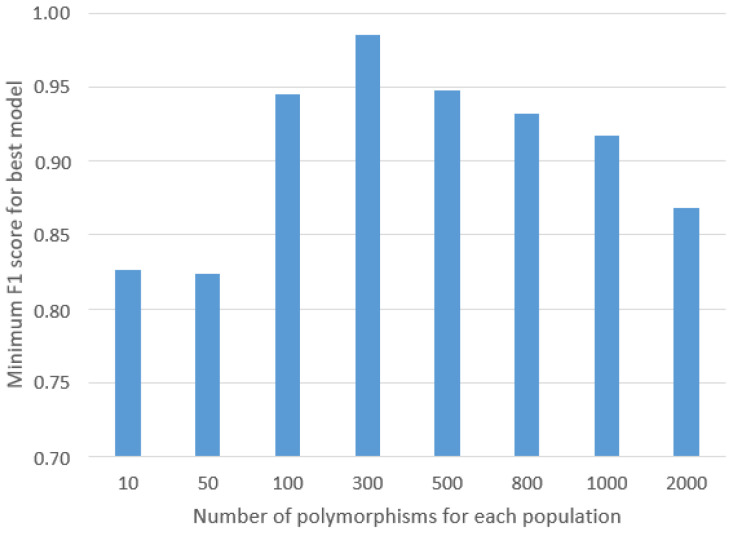
Polymorphism number evaluation histogram, based on GA results. As presented in the histogram for the chosen classifiers, the best results were obtained for 300 SNPs. A further increment resulted in the evaluation metrics’ decrement.

**Figure 2 ijms-24-15095-f002:**
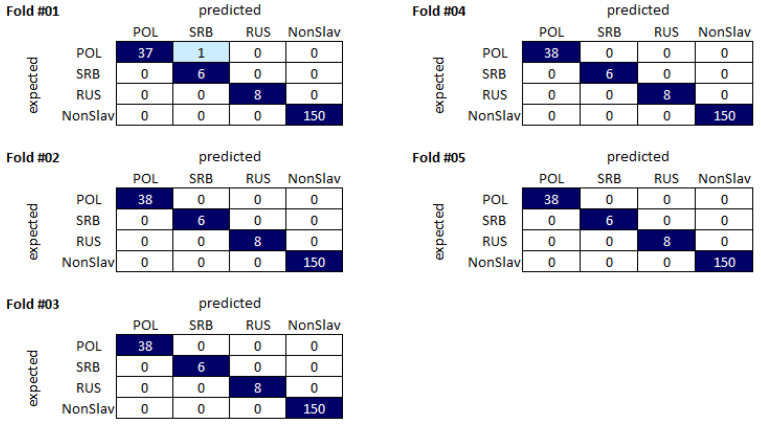
Confusion matrices for each fold for the most-promising classifier. The presented confusion matrices illustrate the aggregated results across all five folds of the experiment. The matrix provides a comprehensive overview of the classification outcomes for all four classes, aiding in the assessment of the model’s performance across different scenarios.

**Figure 3 ijms-24-15095-f003:**
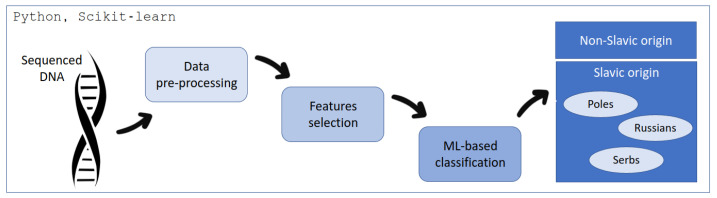
The pipeline of the proposed method. In order to prepare exome data in a form suitable for machine learning (ML) methods, the data were subjected to the pre-processing and feature-selection stage, which was aimed at selecting the key SNP classification.

**Figure 4 ijms-24-15095-f004:**
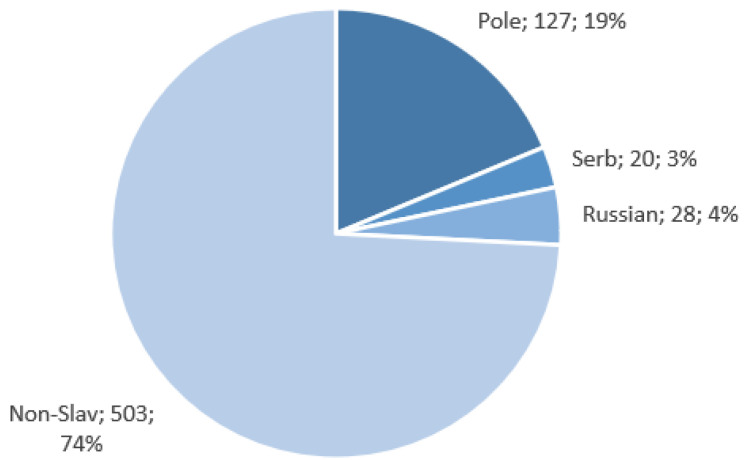
Nationality distribution of the data. Three Slavic populations were tested along with a non-Slavic group, consisting of samples of European origin from The 1000 Genomes Project [7].

**Figure 5 ijms-24-15095-f005:**
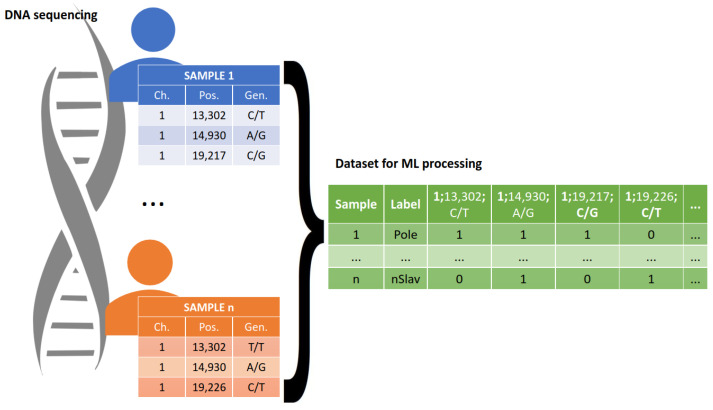
Data pre-processing scheme. Exome sequence from each individual was transformed into a summary table, where each sample is represented in one row. Each column represents the SNP and genotype. If there is such a genotype in a given SNP, the value of 1 is inserted in the summary table. When there is no such case, the value of 0 is inserted instead.

**Table 1 ijms-24-15095-t001:** The results obtained for the selected classifiers. Presented results are the mean values acquired for each classifier on the test dataset. The most-promising results are highlighted in bold.

Classifier	Accuracy	F1-Score			
		**Poles**	**Serbs**	**Russians**	**Non-Slavs**
Linear SVM	0.9990	0.9974	0.9846	1.0000	1.0000
Polynomial SVM	0.9815	0.9581	0.6267	0.9684	0.9987
RBF SVM	0.9941	0.9851	0.8752	1.0000	1.0000
Sigmoid SVM	0.9902	0.9747	0.8108	1.0000	1.0000
Random Forest	0.9776	0.9439	0.3971	1.0000	1.0000
XGBoost	0.9805	0.9525	0.6362	0.9882	0.9980

## Data Availability

Data collected within the NEXT project are available to the public at Mendeley Data https://data.mendeley.com/datasets/9ffz69hbcf (accessed on 6 October 2023).

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
