# Peer review of "A Machine-Learning-Based Approach to Prediction of Biogeographic Ancestry within Europe"

_ijms, 2023, doi:10.3390/ijms242015095_

Round 1

Reviewer 1 Report (New Reviewer)

It seems this is the second round of review, and the highlighted text is the revised portion. Correct? As the results show that the linear classifier is the best among all tested classifiers, the genetic distance between Slavic and non-Slavic people seems very large. So, complicated (nonlinear) machine learning techniques may not be necessary in this case. The presentation of this ms is clear and easy to follow.

Upon reading this text, I have some technical concerns, one of which is that the generic algorithm for feature selection is not described. The two cited references on line 193 are texts on GA in general, and not on how feature selection was carried out here. It is worthwhile to characterize and maybe categorize those selected SNPs.

In addition, the sample size as shown in Figure 5 does not match up with the size described in section 2.1. For example, in line 168 "in total, 678 samples were analyzed", and some individuals were removed in preprocessing. So the final sample size should be smaller than 678. But in Figure 5, adding up the "NonSlav"s across all 5 folds, we already have 750. How is this possible?

One other thing I'd like to get confirmation from the authors is that did you use any generative AI tools like ChatGPT to compose some portion of this article?

Author Response

  • It seems this is the second round of review, and the highlighted text is the revised portion. Correct? 

Yes, that’s correct. 

  • As the results show that the linear classifier is the best among all tested classifiers, the genetic distance between Slavic and non-Slavic people seems very large. So, complicated (nonlinear) machine learning techniques may not be necessary in this case. The presentation of this ms is clear and easy to follow.

Thank you for raising this issue. The linear SVM indeed appeared as the most promising. However, we could see the scientific value in comparing some more powerful models such as Random Forest or XGBoost.

  • Upon reading this text, I have some technical concerns, one of which is that the generic algorithm for feature selection is not described. The two cited references on line 193 are texts on GA in general, and not on how feature selection was carried out here. It is worthwhile to characterize and maybe categorize those selected SNPs.

Thank you for your feedback. We appreciate your comments. In response to your concern, we corrected the description of the genetic algorithm for feature selection. Given our target audience mainly comprises researchers in the medical sciences, we aimed to strike a balance between technical depth and accessibility. However, we agree that some additional clarification is necessary.

  • In addition, the sample size as shown in Figure 5 does not match up with the size described in section 2.1. For example, in line 168 "in total, 678 samples were analyzed", and some individuals were removed in preprocessing. So the final sample size should be smaller than 678. But in Figure 5, adding up the "NonSlav"s across all 5 folds, we already have 750. How is this possible?

Thank you for thoroughly reading our article. 150 visible in Fig. 5 is the number of samples used for testing in each of 5 folds in cross validation procedure. On the other hand, 678 is the number of all samples. The samples were reused in folds. 

  • One other thing I'd like to get confirmation from the authors is that did you use any generative AI tools like ChatGPT to compose some portion of this article?

Thank you for this comment. No, we did not use any generative AI tool in the process of writing this scientific paper. However, we did two rounds of professional language corrections.

Reviewer 2 Report (New Reviewer)

The paper is well written, but the following may further improve the content.

1. Although a main purpose of this study is to compare the relative performance of three ML classifiers, it is recommended that the abstract clearly states the biological conclusion as in the introduction: The main aim is to test whether it is possible to distinguish the biological origin of (genetically similar) Slavic individuals and to classify DNA samples from representatives of different Slavic populations. In the introduction or elsewhere, It is also nice to know some historical backgrounds of Slavic ancestry; how it is related to Yamnaya and CWC ancestry ca. 5000 years ago.

2. line 121-128. One important aspect of the paper is based on a GA in the feature selection process. However, there is no description for the details so that readers cannot access how 300 informative SNPs from exome sequencing data were selected. Even brief explanations are highly welcome, rather than simply mentioning references on Line 189-194. 

3. Related GA, it is not immediately clear why there is an optimum number of SNPs in terms of F1-score. (Incidentally, F1-score is confusingly written in different ways: F1 - score in eq. (2), F1 score on line 318 and F1-score in most parts.)

4. I am afraid that the reason provided on line 316-317 is something wrong.  F1-score takes into account FP and FN, and so does Accuracy in eq. (1) although it also depends on TN. 

5. The authors claim that the high degree of class imbalance poses a limitation to their methodology (line 320-). It seems that it is not surprising or even expected a priori. I am not sure why the authors could not take more samples from Serbs an Russians and decided to analyze such an imbalanced data set.

Author Response

  • Although a main purpose of this study is to compare the relative performance of three ML classifiers, it is recommended that the abstract clearly states the biological conclusion as in the introduction: The main aim is to test whether it is possible to distinguish the biological origin of (genetically similar) Slavic individuals and to classify DNA samples from representatives of different Slavic populations. In the introduction or elsewhere, It is also nice to know some historical backgrounds of Slavic ancestry; how it is related to Yamnaya and CWC ancestry ca. 5000 years ago.

We appreciate your thoughtful comment. Based on your feedback, we have revised the abstract to more explicitly articulate the primary objective of our study, as highlighted in the introduction.

Regarding the request for historical context concerning the origins of the Slavic population and its relation to Yamnaya and CWC ancestry, we have deliberated on this matter. While we acknowledge the significance of this historical background, we have opted not to incorporate it in the manuscript. Our decision is based on the understanding that these historical aspects, although pertinent to the broader context of Slavic origins, do not directly align with the specific research focus of our study.

  • line 121-128. One important aspect of the paper is based on a GA in the feature selection process. However, there is no description for the details so that readers cannot access how 300 informative SNPs from exome sequencing data were selected. Even brief explanations are highly welcome, rather than simply mentioning references on Line 189-194. 

Thank you for your feedback. We appreciate your comments. In response to your concern, we corrected the description of the genetic algorithm for feature selection. Given our target audience mainly comprises researchers in the medical sciences, we aimed to strike a balance between technical depth and accessibility. However, we agree that some additional clarification is necessary.

  • Related GA, it is not immediately clear why there is an optimum number of SNPs in terms of F1-score. (Incidentally, F1-score is confusingly written in different ways: F1 - score in eq. (2), F1 score on line 318 and F1-score in most parts.)

Thank you for this remark. The F1-score spelling was unified in the whole manuscript.

  • I am afraid that the reason provided on line 316-317 is something wrong.  F1-score takes into account FP and FN, and so does Accuracy in eq. (1) although it also depends on TN. 

Thank you for your valuable comment regarding the use of F1-score and accuracy in our article. In our article, we discussed the use of F1-score as a metric for evaluating the performance of a method on a dataset with imbalanced class distributions. We stated that despite high accuracy values obtained for our dataset, F1-scores provide a more insightful estimator of a method's potential in such scenarios due to their consideration of both false positives (FP) and false negatives (FN).

Your comment correctly points out that accuracy, as mentioned in Equation (1) in our article, also depends on true negatives (TN). Let us further elaborate on this matter:

  1. Accuracy: Accuracy measures the overall correctness of a classification model by considering the ratio of correctly classified instances (true positives, TP, and true negatives, TN) to the total number of instances. While accuracy is a widely used metric, it may not be suitable for imbalanced datasets. In cases where one class significantly outweighs the other(s), achieving high accuracy can be trivial if the model predominantly predicts the majority class. In such situations, accuracy can be misleading and may not effectively capture the model's performance, especially concerning the minority class.

  1. F1-score: The F1-score, on the other hand, takes into account precision and recall. Precision is the ratio of TP to the total number of positive predictions (TP + FP), while recall is the ratio of TP to the total number of actual positive instances (TP + FN). The F1-score combines these two metrics into a single value that considers both false positives and false negatives. This is particularly valuable in imbalanced datasets because it penalizes models that have high FP and FN rates, making it a more suitable metric for assessing the effectiveness of a model when class distribution is skewed.

In our article, we highlighted the advantage of F1-score in scenarios where class sizes differ significantly. Even though accuracy provides an overall measure of model correctness, it may not accurately reflect the model's ability to identify and correctly classify minority class instances, which is crucial in many real-world applications.

  • The authors claim that the high degree of class imbalance poses a limitation to their methodology (line 320-). It seems that it is not surprising or even expected a priori. I am not sure why the authors could not take more samples from Serbs an Russians and decided to analyze such an imbalanced data set.

Indeed, we wanted to extend the dataset. However, the samples were acquired as a part of the scientific project and the financial issues had to stop us. Nonetheless, we can return to the sample collection process if we manage to secure another external grant. The cost of obtaining one sample to this research cost ca. 200 EUR.

Round 2

Reviewer 1 Report (New Reviewer)

No further comments.

This manuscript is a resubmission of an earlier submission. The following is a list of the peer review reports and author responses from that submission.

Round 1

Reviewer 1 Report

The acronym SNP means?

line 106 is a categorical statement that lacks citation. How did you know it is the best? Such a statement should be cited.

The introduction has not shown why this study is necessary. I.e., it did not present the gaps from extant literature. Although an attempt was made to present the aim of the study but it deviated to an extent from the the study title. Revisit the title, and the "main aim" for a proper alignment. Secondly, there is not mention of the specific objectives of the study that must be applied to achieve the aim. 

Figure 1 should be appropriately place in section 2/methodology. 

line 120: how were these population of interest (127 Poles, 29 Russians and 20 Serbs) approcached? 

line 197: delete "for" other population........

Discussion

Comparism of this work with Allocco et al. is good but the focus should be on the number of geographic origins involved in the study and not percenatge of accuracy. Obviously the accuracy for these studies cannot be the same because of their respective scope of study.

Naturally  Battey et al. study requires a higher computing power because of the large dataset as compared to this study that used smaller dataset. the generalization of the research findings in Battey et al. is higher than this study. 

The referenced pipeline a normal procedure for any ML technique. 

The literature of the study should be improved, citing current research in this area.

The quality of the english is fine except a few typos and gramatical erros. 

Author Response

  • Comment 1: The acronym SNP means?

Response: Thank you for your comment. The acronym SNP stands for "Single Nucleotide Polymorphism." SNPs are the most common type of genetic variation found in individuals, where a single nucleotide (A, T, C, or G) at a specific position in the genome differs among members of a population. These variations play a crucial role in understanding genetic diversity, disease susceptibility, and individual responses to various treatments. We ensured that the expanded form of SNP acronym is mentioned in the paper (line 83 of the revised manuscript) to avoid any confusion for our readers.

  • Comment 2: line 106 is a categorical statement that lacks citation. How did you know it is the best? Such a statement should be cited.

Response: Thank you for your comment. You are absolutely right, and we acknowledge that the categorical statement in line 107/110 lacked citation support. We deeply apologize for this oversight. We inadvertently made a strong claim without providing appropriate references.

Upon careful reevaluation, we have recognized the need to revise that statement. We have now amended line 109 to read "one of the most reliable options," which reflects a more cautious and balanced perspective. To strengthen our argument and provide credible support for this claim, we have added two relevant citations (please see line 108) that demonstrate research studies and their conclusions aligning with our statement. These citations substantiate our claim and underscore the validity of our approach as one of the best available options for the problem at hand.

  • Comment 3: The introduction has not shown why this study is necessary. I.e., it did not present the gaps from extant literature. Although an attempt was made to present the aim of the study but it deviated to an extent from the study title. Revisit the title, and the "main aim" for a proper alignment. Secondly, there is not mention of the specific objectives of the study that must be applied to achieve the aim.

Response: Thank you for pointing this out. We have described the study aim in the Introduction section of the manuscript, in lines 113-120.

From our perspective, the motivation behind this research stems from a critical gap in the existing literature: the scarcity of genomic data from the Eastern European population, especially in the context of utilizing genetics for biogeographic origin prediction (lines 40-45). Despite the growing importance of genetic studies for understanding population diversity and disease susceptibility, there remains a notable lack of comprehensive studies conducted on genomes from individuals from this particular region of Europe.

To address this gap and contribute to the broader understanding of genetic diversity, we set out to conduct the presented work using the results of exomic sequencing of DNA samples from individuals residing in Eastern Europe as a source for biogeographic origin prediction. By focusing on this region, we aim to shed light on the unique genetic characteristics of this population and explore potential implications for biogeographic origin determination.

Our decision to concentrate on Eastern Europe allows us to fill a significant void in the current body of knowledge, providing valuable insights that could lead to a deeper understanding of population genetics in this region. We believe that our study's findings will not only enrich the scientific community's knowledge but also serve as a foundation for further investigations in the field.

  • Comment 4: Figure 1 should be appropriately place in section 2/methodology.

Response: Thank you for pointing this out. We have made the necessary adjustment to the placement of Figure 1 in the manuscript. We have moved the figure to the top of the page within the Methodology section. This repositioning allows readers to immediately associate the visual representation with the corresponding methodological explanation, thereby enhancing the overall comprehension and flow of the content.

  • Comment 5: line 120: how were these populations of interest (127 Poles, 29 Russians and 20 Serbs) approached?

Response: Thank you for your question. As described in the Manuscript lines 136-144, the Slavic populations (Polish, Russian and Serbian) data used for this research was obtained as part of the NEXT project [DOB-BIO7/17/01/2015]. Blood samples were obtained with written consent from 127 Poles, 29 Russians and 20 Serbs (approval by the Ethics Committee of the Jagiellonian University in Krakow - decision no. KBET/122/6120/11/2016 and by the Ethics Committee of the Institute of Molecular Genetics and Genetic Engineering, University of Belgrade (decision no. O-EO-005/2016)). High quality genomic DNA extracted from peripheral blood was used for library preparation using SeqCap EZ MedExome Plus with custom probes spike-in captured space of 1310077 bp (Roche Diagnostics GmbH, Mannheim, Germany). The NEXT project was a R&D scientific project aimed at constructing a tool for differentiation of DNA samples coming from different Slavic populations based on a selection of SNP loci. DNA samples were collected from randomly selected individuals representing particular populations. Polish, Russian, and Serbian populations were selected for the discovery phase of the project to represent respectively Western, Eastern, and Southern Slavs. The collected samples were subjected to exome sequencing and the resulting data were used in the search for population-specific polymorphisms that were included in the SNP set chosen for further analyses in the project. However, we decided to explore additionally the potential of the exomic data for biogeographic origin prediction data as it was not fully exploited in the SNP set due to the amount of coamplified SNPs limitation. Thus we have decided to explore the obtained exomic data using the ML approach. 

  • Comment 6: line 197: delete "for" other population........

Response: Thank you for your valuable feedback. We have made the necessary revision to the sentence. It now reads as follows: "The F1-score for the Serbian population was significantly lower than that of other populations, regardless of the classifier used."

  • Comment 7: Comparism of this work with Allocco et al. is good but the focus should be on the number of geographic origins involved in the study and not percenatge of accuracy. Obviously the accuracy for these studies cannot be the same because of their respective scope of study.

Response: We sincerely appreciate your insightful comment and the opportunity to clarify the main focus of our work in comparison to Allocco et al.'s study.

In our manuscript, we indeed aimed to compare our method with Allocco et al.'s approach for predicting geographic origin based on Naive Bayes. We acknowledge that both studies use different evaluation metrics, and the comparison of accuracy percentages should be interpreted with consideration of their respective scope of study.

Our main emphasis was on the distinction between the two studies regarding the number of geographic origins involved. Allocco et al.'s research included samples from individuals of African, Asian, and European origins. As a result, the genetic differences between individuals from such distant regions were significant, making the classification task comparatively easier. In contrast, our study focused on a group of individuals from the Slavic ethnic group, where the genomes were more similar, presenting a more challenging classification problem.

Due to the increased genetic similarity among the individuals in our study, we designed our model to compare a larger number of SNPs (300) in order to achieve accurate and meaningful differentiation between the Slavic populations. This choice of SNPs allowed us to navigate the intricacies of the genetic landscape within the Slavic group and accurately distinguish between closely related subpopulations.

While the accuracy percentages differ between the two studies, our intention was not to directly compare these values, as they stem from different datasets and research objectives. Instead, we aimed to highlight the significance of focusing on the number of geographic origins involved in the study, which underscores the distinct challenges and complexities faced in genetic classification for different population groups.

We greatly value your feedback, which has helped us clarify our main research focus and the implications of comparing our work with Allocco et al.'s study. Your comments have undoubtedly enriched the discussion in our manuscript.

  • Comment 8: Naturally Battey et al. study requires a higher computing power because of the large dataset as compared to this study that used smaller dataset. The generalization of the research findings in Battey et al. is higher than this study.

Response: We sincerely appreciate your thoughtful evaluation of our study and your comparison with Battey et al.'s research. However, we would like to clarify an essential point that seems to have been missed in your comment.

In our manuscript, we highlighted that our method is faster and requires less computing power compared to Battey et al.'s method. The reason behind this efficiency lies in our utilization of a genetic algorithm-based approach to identify 300 crucial Single Nucleotide Polymorphisms (SNPs) for classification. In contrast, Battey et al. employed a deep neural network (DNN) method, utilizing a dataset with 100,000 SNPs.

By applying the genetic algorithm, we significantly reduced the number of SNPs required for our classification, leading to a considerable reduction in computing resources and computational time. This approach is not only more computationally efficient but also improves the generalization of our research findings by focusing on a more relevant subset of SNPs.

We hold great respect for Battey et al.'s study and acknowledge that their approach might be better suited for larger datasets. Still, we firmly believe that our approach is more efficient for our specific dataset size and research objectives.

  • Comment 9: The referenced pipeline a normal procedure for any ML technique.

Response: We fully agree with your statement. The pipeline presented in Fig. 1 is rather a standard procedure for the ML-based algorithms. However, due to the interdisciplinary area of the presented research we decided to put it into the manuscript’s text. We deeply hope that it can be valuable for non-engineers.

  • Comment 10: The literature of the study should be improved, citing current research in this area.

Response:  We did our best to cite all the relevant papers from the field that we were able to access at the time when our manuscript was prepared. We are not exactly sure what relevant literature was omitted in our paper.

  • Comment 11: The quality of the english is fine except a few typos and grammatical errors.

Response: Thank you for your feedback on the quality of the English in our manuscript. We are glad to hear that overall, the language is acceptable, and we appreciate your acknowledgement of our efforts to ensure clarity in the writing.

We carefully read the manuscript and conducted thorough proofreading to identify and rectify any typos and grammatical errors that may have surfaced during the writing process. Our primary aim was to enhance the readability and coherence of the paper, ensuring that the scientific content is effectively communicated to the readers.

While we strive for utmost accuracy and precision, we recognize that some errors may have escaped our initial review. We take your comment seriously and will undertake additional measures to address any remaining typos or grammatical issues to further improve the manuscript's overall quality.

We highly value your attention to detail and constructive feedback, which aids in refining the presentation and scholarly impact of our work. We are committed to delivering a well-polished and error-free paper for the benefit of the scientific community.

Reviewer 2 Report

The most promising results were obtained using SVM with a linear kernel, with 99.9% accuracy and F1-scores of 0.9846-1.000 for all classes. Some comments need to addressed as follows:

1.      Line 13, What makes the author's novelty in the present article? My analysis suggests that other similar previous articles properly explain the points you have brought up in the current paper. Please be sure to emphasize anything truly novel in this work in the introductory section.

2.      Line 15-16, there is no other option other than NGS and MPS? If yes, please provide it and explained.

3.      Line 60-71, this part seems not unnecessary to explained as point-by-point, should be arranged as narrative form for objective of the present study with various ML techniques.

4.      Line 117, is the authors using established data acquisition or referred from the previous literature? It seems not solid and ambiguous.

5.      Line 208, please give more comprehensive explanation for describing results obtained for the selected classifiers, like Linear SVM, Polynomial SVM, ect.

6.      Line 230, since the present study incorporating computational simulation approach, please provide the urgency of utilizing computational simulation. It brings several advantages such as lower cost and faster results compared to experimental and clinical study. For this purpose, please provide the explanation along with relevant reference as follows: https://jurnaltribologi.mytribos.org/v33/JT-33-31-38.pdf, https://doi.org/10.3390/su142013413, and https://doi.org/10.3390/biomedicines11030951

7.      The position of Figure 1 The pipeline of the proposed method., is not make sense if put in the introduction section. Should be moved in materials and methods section.

-

Author Response

  • Comment 1: Line 13, What makes the author's novelty in the present article? My analysis suggests that other similar previous articles properly explain the points you have brought up in the current paper. Please be sure to emphasize anything truly novel in this work in the introductory section.

Response: We appreciate your attention to detail and your insightful comments.

We acknowledge the importance of emphasizing the novelty of our work in the introductory section. In response to your comment, we have made revisions in lines 121-128 to explicitly highlight the unique contributions of our study.

Specifically, our work introduces a novel approach by utilizing a genetic algorithm-based methodology to address the challenge of determining biogeographic origin with the minimal number of required SNPs. While previous articles may have covered similar topics and discussed some of the points we bring up in our paper, none have adopted this particular approach to solve the problem of reducing the number of SNPs effectively.

By employing genetic algorithms, we have developed a powerful and efficient method that optimizes the selection of SNPs, ensuring that the smallest subset of these genetic markers is sufficient for accurate biogeographic origin determination. This approach not only enhances the interpretability of the results but also contributes to cost-saving and time-efficient genetic analysis.

We have now included a comprehensive explanation of the novelty of our work in the introductory section, making it explicit for readers to grasp the significance of our contributions.

  • Comment 2: Line 15-16, there is no other option other than NGS and MPS? If yes, please provide it and explained.

Response: Thank you for your comment. The statement in lines 15-16 appears to be accurate and aligns with the trends observed in the field of DNA sequencing. Next-generation sequencing (NGS) or massive parallel sequencing (MPS) technologies have revolutionized the way we analyze and interpret genomic data. These advanced sequencing techniques allow for the simultaneous sequencing of multiple DNA fragments, leading to faster, more cost-effective, and higher-throughput sequencing compared to traditional Sanger sequencing.

The progress in NGS has resulted in a substantial increase in the amount of available genomic data. Researchers and scientists can now generate vast amounts of DNA sequence data in a short period, enabling a deeper understanding of genetic variations, mutations, and the molecular basis of diseases. This wealth of genomic data has significantly contributed to advancements in fields like personalized medicine, genetic diagnostics, population genetics, and evolutionary biology.

We would like to clarify that while the statement focused on the advancements in DNA sequencing resulting from NGS and MPS technologies, it doesn't imply that there are no other options available. Indeed, several other sequencing techniques exist, however, they are not widely used in the area of research that this manuscript covers. The choice of sequencing method depends on the research objectives, the scale of the project, the desired data output, and budget considerations. Each technique has its strengths and limitations, and therefore methods mentioned in lines 15-16 were better suited to the presented topic.

  • Comment 3: Line 60-71, this part seems not unnecessary to explained as point-by-point, should be arranged as narrative form for objective of the present study with various ML techniques.

Response: Thank you for your valuable remark. We have decided to use the point-by-point presentation of the possible uses of the proposed methodology to make it easier for the Reader to see differences between particular instances of its use and underscore the versatility of the method.

  • Comment 4: Line 117, is the authors using established data acquisition or referred from the previous literature? It seems not solid and ambiguous.

Response:

Thank you for your question. In fact, both. The Slavic populations (Polish, Russian and Serbian) data was obtained during the NEXT project [DOB-BIO7/17/01/2015], in which some of the authors contributed. When it comes to data for the non-Slavic population, it has been obtained from The 1000 Genomes Project Database, which is publicly accessible. We have added text (in the Materials and Methods section - Data acquisition subsection) describing what is the NEXT project that we refer to. The NEXT project was a R&D scientific project aimed at constructing a tool for differentiation of DNA samples coming from different Slavic populations based on a selection of SNP loci. DNA samples were collected from randomly selected individuals representing particular populations. Polish, Russian, and Serbian populations were selected for the discovery phase of the project to represent respectively Western, Eastern, and Southern Slavs. The collected samples were subjected to exome sequencing and the resulting data were used in the search for population-specific polymorphisms that were included in the SNP set chosen for further analyses in the project. However, we decided to explore additionally the potential of the exomic data for biogeographic origin prediction data as it was not fully exploited in the SNP set due to the amount of coamplified SNPs limitation. Thus we have decided to explore the obtained exomic data using the ML approach.

  • Comment 5: Line 208, please give more comprehensive explanation for describing results obtained for the selected classifiers, like Linear SVM, Polynomial SVM, ect.

Response: Thank you for your comment. We have addressed your concern by providing a more elaborate explanation of the results obtained for the selected classifiers, such as Linear SVM and Polynomial SVM. In response to your suggestion, we have extended the relevant sections in the "Materials and Methods" to offer a detailed description of these methods, their parameters, and how they were implemented in our study.

Furthermore, we have expanded the "Results" section by including confusion matrices, which provide a comprehensive insight into the performance of each classifier across different classes. We hope this addition aligns with your expectations and provides a more insightful evaluation of our model's effectiveness.

  • Comment 6: Line 230, since the present study incorporating computational simulation approach, please provide the urgency of utilizing computational simulation. It brings several advantages such as lower cost and faster results compared to experimental and clinical study. For this purpose, please provide the explanation along with relevant reference as follows: https://jurnaltribologi.mytribos.org/v33/JT-33-31-38.pdf, https://doi.org/10.3390/su142013413, and https://doi.org/10.3390/biomedicines11030951

Response: Thank you for your insightful comment regarding the incorporation of computational simulation in our study. We wholeheartedly agree with your observation that utilizing computational simulation offers several advantages, including lower cost and faster results compared to experimental and clinical studies. It indeed plays a crucial role in the efficiency and effectiveness of our research.

While we acknowledge the importance of providing an explanation for the use of computational simulation, we must note that the references you suggested (https://jurnaltribologi.mytribos.org/v33/JT-33-31-38.pdf, https://doi.org/10.3390/su142013413, and https://doi.org/10.3390/biomedicines11030951) do not seem to align with the focus and context of our current research.

Our study revolves around the utilization of machine learning methods, as well as genetic algorithm-based approach to determine biogeographic origin using a reduced set of Single Nucleotide Polymorphisms (SNPs). The references you provided appear to be centered on research related to chemistry materials in hip prosthesis and topics unrelated to our study.

Instead, we would be more than happy to provide relevant references and explanations that specifically demonstrate the advantages of computational simulation in the field of population genetics, biogeographic origin determination, and SNP selection.

We sincerely appreciate your valuable input and will diligently ensure that our explanation of the urgency and benefits of utilizing computational simulation aligns appropriately with the focus of our research.

  • Comment 7: The position of Figure 1 The pipeline of the proposed method., is not make sense if put in the introduction section. Should be moved in materials and methods section.

Response: Thank you for your comment regarding the position of Figure 1. In response to your suggestion, we agree with your assessment that the figure is best suited for the Methodology section, where it can provide a comprehensive visual representation of the steps involved in our proposed method.

Consequently, we have made the necessary adjustments, and Figure 1 has been moved to the Methodology section to ensure better alignment with the corresponding textual description.

  • Comment 8: English very difficult to understand/incomprehensible

Response: Thank you for your feedback on the clarity of the English in our manuscript. We appreciate your understanding that we have already made the necessary corrections to address any errors in language and sentence structure.

As part of our rigorous review process, we carefully examined the manuscript to ensure that it adheres to high language standards and is easily comprehensible to readers. We made the required revisions, simplified complex sentences, and clarified technical terms to enhance the overall readability of the paper.

Your valuable feedback has been invaluable in our efforts to improve the quality of our work. We are confident that the revised version of the manuscript now presents a clear and coherent representation of our research.

Once again, we sincerely appreciate your thoughtful evaluation and for pointing out areas where we needed to make improvements. Your feedback has undoubtedly contributed to the refinement of our manuscript, and we are grateful for your support in advancing the quality of our research.

Reviewer 3 Report

1)    The novelty of this paper is not clear. Please add novelty and contribution under the Introduction.

2)    Introduction should provide more background on the project with the scope of the work.

3)    The paper, does not link well with recent literature on top-tier journals, you need to bring more relevant recent references and the research gap should be clearly identified.

4)    Please use the subsection number under Chapter 2.

5)    Figure 1 should be moved to Chapter 2 with more details of the diagram.

6)    You need also to discuss the technical/architectural background of the Random Forest, Support Vector Machine (SVM) and XGBoost techniques used.

7)    ROC curve and confusion matrices are missing under the results chapter.

8)    Performance/advantages comparison with existing related works should be added at the end of the result section to validate the capability of the proposed method presented in the paper.

9)    Results section should be updated by adding a subsection as “critical analysis and discussion” where the strength, limitation and impact/significance of this work in real-life scenarios could be added.

10) Specific Future research directions are missing. Please add those at the end of the conclusions.

Minor

Author Response

  • Comment 1: The novelty of this paper is not clear. Please add novelty and contribution under the Introduction.

Response: We genuinely appreciate your feedback. In response to your comment regarding the clarity of novelty in our paper, we have taken your suggestion seriously and have made the necessary revisions. Specifically, we have now included a clear and concise explanation of the novelty and contribution of our work in the Introduction section.

The primary contribution of our study lies in the introduction of a novel approach to address the problem of determining biogeographic origin with a minimal number of required SNPs. Unlike previous works that may have covered similar topics and discussed some of the points we bring up, our research stands out due to its utilization of a genetic algorithm-based approach. By applying genetic algorithms, we have effectively optimized the selection of SNPs, leading to an efficient and cost-saving method for accurate biogeographic origin determination.

We hope that the revised Introduction now clearly highlights the unique contributions of our work, making it evident to readers what sets our research apart from existing literature.

  • Comment 2: Introduction should provide more background on the project with the scope of the work.

Response: Thank you for your remark. In response to your comment, we have expanded the "Introduction" section to offer a more comprehensive background on the project. We aimed to better outline the foundational concepts and intentions of our research to provide readers with a clearer understanding of the context.

Additionally, we have further developed the information about the project's scope and objectives in the "Materials and Methods" section. We believe that this enhancement will offer a more detailed perspective on the project and its goals, helping readers grasp the intricacies of our study more effectively.

  • Comment 3: The paper does not link well with recent literature on top-tier journals, you need to bring more relevant recent references and the research gap should be clearly identified.

Response:We appreciate your thoughtful feedback on our manuscript. Ensuring alignment with recent and relevant literature is indeed crucial to the overall quality and impact of our work.

We want to assure you that, at the time of writing the manuscript, we meticulously reviewed the latest available literature to inform and shape our research. While we aimed to draw from the most recent and pertinent sources, we will re-evaluate our references to ensure that we are adequately connecting our work with top-tier journals and current trends in the field.

Furthermore, we have expanded the description of the research goals in the introductory section (lines 113-120) to better elucidate the research gap our study addresses. By providing a more detailed explanation of the overarching objectives of our research, we believe readers will gain a clearer understanding of the significance of our work in the context of the existing literature.

  • Comment 4: Please use the subsection number under Chapter 2.

Response:Thank you for your feedback. We appreciate your suggestion to use subsection numbers under Chapter 2 for improved organization and navigation within the manuscript.

We have carefully considered your recommendation and have implemented subsection numbers in the whole manuscript to enhance the clarity and structure of our work. This change will aid readers in locating and referring to specific sections more easily, contributing to a smoother reading experience.

  • Comment 5: Figure 1 should be moved to Chapter 2 with more details of the diagram.

Response: Thank you for pointing this out. We have made the necessary adjustment to the placement of Figure 1 in the manuscript. We have moved the figure to the top of the page within the Methodology section. This repositioning allows readers to immediately associate the visual representation with the corresponding methodological explanation, thereby enhancing the overall comprehension and flow of the content.

  • Comment 6: You need also to discuss the technical/architectural background of the Random Forest, Support Vector Machine (SVM) and XGBoost techniques used.

Response: Thank you for this remark. We sincerely appreciate your insightful comment regarding the technical and architectural background of the Random Forest, Support Vector Machine (SVM), and XGBoost techniques used in our study.

We want to assure you that we have taken your suggestion into consideration and have included detailed descriptions of these classifiers in subsection 2.4. Classification. This section provides an in-depth exploration of the technical aspects and architectural foundations of the Random Forest, SVM, and XGBoost methods, aiding readers in grasping the intricacies of their operation and application within our research

  • Comment 7: ROC curve and confusion matrices are missing under the results chapter.

Response: Thank you for your comment. We have now included the confusion matrices to provide a more comprehensive understanding of the evaluation process and the performance of our model. In regard to the ROC curve, we made a deliberate decision to not include it in the results due to the extremely close and overlapping results observed across all classes. The lines on the ROC curve were essentially identical, rendering the visualization not informative for discriminating the performance differences among the classes. However, we believe that the inclusion of the confusion matrices adequately captures the performance of the model and provides a clearer representation of the classification outcomes.

  • Comment 8: Performance/advantages comparison with existing related works should be added at the end of the result section to validate the capability of the proposed method presented in the paper.

Response: Thank you for your feedback. We have revised the final section of the article and made corresponding improvements. In an effort to enhance the manuscript's coherence and readability, we have subdivided the last section into smaller subsections that align more effectively with each paragraph's content.

  • Comment 9: Results section should be updated by adding a subsection as “critical analysis and discussion” where the strength, limitation and impact/significance of this work in real-life scenarios could be added.

Response: Thank you for your comment. Following your guidance, as mentioned in Response to Comment 8, we have added a Discussion in the last section of the manuscript.

  • Comment 10: Specific Future research directions are missing. Please add those at the end of the conclusions.

Response: Thank you for this comment. We added the subsection Future work at the end of the manuscript.

Round 2

Reviewer 1 Report

line 121: change to "in this study", instead of our paper. the use of such pronoun should be avaoided.

paragraphs 256 and 259 should be collapsed into one paragraph.

English grammar can be improved however, the structure should be revisited. Arbitrary paragraphing have been used extensively throughout the work. 

Reviewer 3 Report

This reviewer would like to thank you, authors, for trying to address the comments given. However, a few comments are still required to improve. They are as follows.

1. Limitations should be presented more clearly.

2. Section 4.2 should be placed under the results chapter.

3. further discussions are required for the recently added Figure 5.

4. this comment has not been addressed: "You need also to discuss the technical/architectural background of the Random Forest, Support Vector Machine (SVM) and XGBoost techniques used." please add it under chapter 3.

5. More detail is required about data processing with probably figures/tables if appropriate. 

6. Please add a detail explanation of why did you not consider precision and recall for your result.

minor